# Switch of Nocturnal Non-Invasive Positive Pressure Ventilation (NPPV) in Obstructive Sleep Apnea (OSA)

**DOI:** 10.3390/jcm11113157

**Published:** 2022-06-01

**Authors:** Pasquale Tondo, Caterina Pronzato, Irene Risi, Nadia D’Artavilla Lupo, Rossella Trentin, Simona Arcovio, Francesco Fanfulla

**Affiliations:** 1Respiratory Function and Sleep Medicine Unit, Istituti Clinici Scientifici Maugeri IRCCS, 27100 Pavia, Italy; pasquale.tondo@unifg.it (P.T.); caterina.pronzato@icsmaugeri.it (C.P.); irene.risi@icsmaugeri.it (I.R.); nadia.dartavillalupo@icsmaugeri.it (N.D.L.); rossella.trentin@icsmaugeri.it (R.T.); simona.arcovio@icsmaugeri.it (S.A.); 2Department of Medical and Surgical Sciences, University of Foggia, 71122 Foggia, Italy

**Keywords:** auto-BiPAP, CPAP, sleep apnea, treatment, VAuto

## Abstract

Background. Continuous positive airway pressure (CPAP) is considered the first-line treatment for patients with OSA, but Bilevel-PAP (BiPAP) therapy is a recognized option for noncompliant/unresponsive patients to CPAP. The present study was designed to evaluate the role of ResMed VAuto in the management of two different issues raised because of the Philips recall: the treatment of naïve noncompliant/unresponsive patients to CPAP (Group A) and the switch to VAuto for patients already on treatment with Philips Auto-BiPAP (Group B). Methods. Sixty-four patients who required auto-BiPAP treatment from August to December 2021 were included in the study. The efficacy of each mode of PAP therapy was compared between the two groups of patients. Results. Group A showed a statistically significant improvement in the apnea–hypopnea index (AHI) (7.4 ± 8.5 events·h^−1^ vs. 15.2 ± 12.1 events·h^−1^, *p* < 0.001), and oxygen desaturation index (ODI) (9.4 ± 8.9 events·h^−1^ vs. 15.2 ± 8.8 events·h^−1^, *p* = 0.029) during VAuto in comparison to CPAP, respectively. Conversely, a similar trend was found for patients in Group B for global AHI, but a statistically significant reduction was just found in supine AHI and ODI. In group B, an AHI <5 events·h^−1^ was found in 89.3% during VAuto in comparison to 82.1% with Philips Auto-BiPAP (*p* = ns). The levels of IPAPmax and EPAPmin were not statistically different between the two devices (*p =* 0.69 and *p =* 0.36, respectively). Conclusion. Bilevel ventilation in VAuto mode is effective in the clinical management of two different issues derived from the Philips recall. The switching between two different auto-BiPAP devices can be performed easily and successfully.

## 1. Introduction

Obstructive sleep apnea (OSA) is a common sleep-related breathing disorder (SRBD) that affects a large number of individuals [1] and is characterized by pharyngeal collapse that can be complete or partial [2]. This nocturnal disorder has an important impact on health by increasing the risk of comorbidities from various causes and mortality [3].

Continuous positive airway pressure (CPAP) is considered the first-line treatment, at least for patients with moderate-to-severe OSA or associated excessive daytime sleepiness [4], but in patients with poor tolerance or poor adherence to CPAP, the switch to bilevel positive airway pressure (BiPAP) therapy is a widely used option worldwide [5,6].

Recently, Philips Respironics Inc (Murrysville, PA, USA) has issued a worldwide warning about the risk of degradation of polyurethane foam (PUF) of several PAP devices used for the long-term treatment of patients with SRBD or Chronic Respiratory Failure [7]. The degraded foam particles may cause airway irritation, as well as the volatile organic components (VOC) released during the degradation process may have carcinogenic effects [8].

This recall provoked serious concerns in patients, physicians, and health care organizations. Moreover, while the simplest form of PAP therapy (CPAP or BiPAP) in OSA patients was easily managed, the auto-BiPAP devices are generally considered as not-exchangeable since a great difference exists in the functioning algorithm between manufacturers.

ResMed AirCurve™ 10 VAuto (ResMed, San Diego, CA, USA) has been proposed for the treatment of severe OSA patients that usually require high-pressure support. Data about the efficacy of this device are limited. Ball et al. reported in a multicenter small size study no differences in efficacy between VAuto and standard in intolerant patients to CPAP [9]. Benjafield et al., in a big data analysis study designed to assess differences in compliance after switching from CPAP to Bilevel therapy, observed no statistical differences in the apnea–hypopnea index (AHI) between CPAP and BiPAP, regardless of the Bilevel-PAP mode (fixed or auto) [6]. However, no data are available on patients that are non-respondent to CPAP.

The present study was designed to evaluate the role of ResMed VAuto in the management of two different issues raised as a consequence of Philips safety recall: the treatment of naïve patients non-respondent or intolerant to CPAP (Group A) and the switch to VAuto for patients already on treatment with Philips Auto-BiPAP (Group B). The primary endpoint was a non-inferiority of VAuto in comparison to Philips Auto-BiPAP expressed as a percentage of patients successfully treated.

## 2. Materials and Methods

Patients that required auto-BiPAP therapy from August to December 2021 were included in the study. A total of 64 patients (70% males, mean age of 67.0 ± 9.5 years) were enrolled: 36 in Group A and 28 in Group B. All patients received OSA diagnosis after clinical evaluation [10] and home full standard polysomnography (PSG_home_). OSA diagnosis was made according to standard criteria; all the patients were eligible for PAP therapy according to international and Italian current guidelines [11]. The study was approved by the Ethical Committee of the Istituti Clinici Scientifici Maugeri IRCCS (n. 2631 CE).

### 2.1. Ventilator Setting

The VAuto mode is an auto-adjusting bilevel device that adjusts the expiratory positive airway pressure (EPAP) in response to flow limitation, snoring, and obstructive sleep apnea. The VAuto device allows you to set a minimum EPAP (EPAP_min_) and a maximum inspiratory positive airway pressure (IPAP_max_) to limit the upper and lower limits of the pressure delivered. The EPAP and IPAP ranges are 4–25 cmH2O, and a fixed level of pressure support (PS) should be set according to patient’s need (maximum level 10 cmH2O) [12].

All the patients performed in-lab full standard polysomnography (NATUS NDx, Natus Medical Inc., Middleton, WI, USA) for VAuto titration. Proper adjustments to ventilator settings were made according to patient’s needs: increasing in EPAP in the case of persistent obstructive apneas; increasing in IPAP_max_ and PS in the case of persistent hypopneas or desaturations.

The titration of ventilator setting was performed with two different approaches according to the two groups of patients:

**Group A.** After a period of acclimatization to PAP and mask, all patients underwent an in-lab manual titration according to standard procedure recommended by AASM [13]. Patients switched to auto-BiPAP if they were not responsive to CPAP or were unable to tolerate CPAP for high pressure. Non-responsive patients were defined according to AASM guidelines: persistence of respiratory disturbance index above the level of optimal, good, or adequate titration. Non-tolerant patients were those with poor tolerance or compliance to CPAP, recognized during the adaptation period using the effective CPAP level. All treatable causes of poor CPAP tolerance or efficacy (type of mask, presence of leaks, poor humidification) were identified and treated or excluded before the switch to auto-BiPAP.

The initial setting of the VAuto was made accordingly to data observed during CPAP titration: the starting EPAP value was the CPAP level that provided correction of obstructive apneas, while the maximum inspiratory pressure was set at 2 cmH2O above the maximum level of CPAP that provided good control of obstructive hypopneas, inspiratory flow-limitation arousals and snoring.

**Group B.** All the patients received a clinical examination to assess presence of symptoms related to device warning. Before the switch to VAuto, patients performed a standard PSG with portable system with the habitual device. The initial setting of VAuto was aimed to reply to the previous physiological setting of Philips device.

### 2.2. Statistical Analysis

Continuous variables were expressed as mean ± standard deviation, whereas categorical variables were expressed as percentage. The main outcome, percentage of patients successfully treated with the two modes of ventilation, and the differences of gender and comorbidities reported by two groups were assessed by the χ^2^ Test. The comparison between the two modes of ventilation was assessed by means of T-test and Z-test for paired samples, separately for each group of patients. The sample size was calculated taking into account the findings of the auto-BiPAP validation study (Carlucci et al., 2015) [5], assuming a variability in success rate of 20% (95% CI, 28.8–43.2) and considering a cut-off level a residual obstructive AHI of 5 events·h^−1^.

Statistical analyses were performed by the IBM SPSS Statistics for Windows (version 26; IBM Corp., Armonk, NY, USA); the graphs were created by GraphPad (version 8; GraphPad Software Inc., San Diego, CA, USA). The significance threshold has been set at *p* < 0.05.

## 3. Results

Table 1 reports anthropometrics and sleep data recorded at diagnosis for all patients.

No differences were found in demographic data and comorbidities prevalence between the two groups of patients, as shown in Table 2.

Table 3 shows the comparison between respiratory parameters recorded during CPAP and VAuto therapy in Group A and during Philips Auto-BiPAP or VAuto therapy in Group B. Overall, 15.6% of patients showed a residual obstructive AHI >5 events·h^−1^ during VAuto ventilation mode.

In Group A, as shown in Figure 1 and Table 3, a statistically significant improvement was found during VAuto ventilation in comparison to CPAP for arousals index (*p <* 0.001), AHI in REM sleep (*p =* 0.002), global AHI (VAuto 7.4 ± 8.5 events·h^−1^ vs. CPAP 15.2 ± 12.1 events·h^−1^, *p* < 0.001), oxygen desaturation index (ODI) (VAuto 9.4 ± 8.9 events·h^−1^ vs. CPAP 15.2 ± 8.8 events·h^−1^, *p* = 0.029) and nadir% (*p =* 0.026). No differences were found between the two modes of ventilation for T_90_ and mean SaO_2_. The percentage of patients successfully treated improved from 38.8 to 80.5% (*p* < 0.01).

A similar trend was found for patients in Group B for global AHI, but a statistically significant reduction was just found in supine AHI and ODI, as reported in Table 2 and Figure 2.

The percentage of responsive patients to VAuto was 89.3% in comparison to 82.1% with Philips Auto-BiPAP (*p* = ns). The level of IPAP_max_ (Philips Auto-BiPAP 17.4 ± 3.3 cmH2O vs. VAuto 17.5 ± 2.9 cmH2O, *p =* 0.69) and EPAP_min_ (Philips Auto-BiPAP 9.4 ± 2.3 cmH2O vs. VAuto 9.5 ± 2.3 cmH2O, *p =* 0.36) were not statistically different between the two devices as shown in Figure 3.

## 4. Discussion

The present study suggests that VAuto is a valuable alternative for patients unresponsive or intolerant to CPAP and that the swap from Philips Auto-BiPAP can be easily performed without substantial changes in the ventilator setting. Furthermore, overall minimal differences in efficacy profile were found between VAuto and Philips devices.

Philips released a Field Safety Notification for PAP devices commonly prescribed for the treatment of SRBD and/or respiratory failure. The notification was based on the risk that volatile gas products or particles from the polyester-based polyurethane foam may be inhaled by the patients during PAP therapy sessions: airway irritation or carcinogenicity were supposed as potential health risks. Consequently, Sleep Laboratories have had to face this new emergency by making the decision to continue or discontinue PAP therapy on the basis of a single patient evaluation.

Patients requiring auto-BiPAP treatment usually showed worse OSA severity, frequent comorbidities, or most pronounced nocturnal gas exchange impairment [14]. However, automatic ventilators are adjusted by algorithms that are quite different among manufacturers, so we moved through uncertainty: maintain therapy with the same device or switch to another.

Data about the VAuto device are very limited. Ball et al. [9] compared in a small sample sized study VAuto and standard fixed BiPAP treatment in a group of intolerant patients to CPAP. They found that VAuto ventilation was better tolerated by the patients, without a clinically significant difference in residual AHI between the two modes. The improvement in tolerance and adherence was subsequently confirmed by Benjafield et al. in a big data analysis study [6].

The present study confirms that VAuto ventilation was well accepted in both intolerant and resistant patients to CPAP. The comparison between VAuto and CPAP showed a marked reduction of AHI (−51.3%) and ODI (−38.2%) during VAuto treatment, demonstrating excellent control of sleep respiratory disturbances and gas exchange abnormalities.

However, clinical data on the use of auto-bilevel PAP are quite limited. Usually, the prescription was secondary to reducing adherence to CPAP, and in almost the studies, a non-inferiority to CPAP treatment was found [14,15,16,17]. The findings of the present study are quite similar to those we previously observed for Philips Auto-BiPAP [5]. However, comparing data of patients who switched from the Philips device to VAuto, we found an improvement in some polysomnographic indices, particularly residual AHI in a supine position and in nocturnal gas exchange. The difference in efficacy could be explained by the differences in the operating algorithm between devices. The EasyBreathe option in the VAuto allows pressure support delivery with a waveform that accompanies the profile of the patient’s inspiratory act, improving patient/ventilator synchrony. Furthermore, in the VAuto device, only EPAP is adjusted according to the presence of residual obstructive events while the level of PS is maintained stable so that we might postulate that VAuto may improve the minute ventilation. On the other hand, since Group B patients were already on treatment with auto-BiPAP for several months or years, we cannot be excluded that the initial setting may be no longer optimal. Indeed, during the nocturnal titration, the VAuto was initially set to reply to those of auto-BiPAP, but adjustments have been made according to the patient’s needs (see Figure 3 for the individual trend) in terms of residual obstructive events or desaturations. No studies are available in the literature comparing two different bilevel auto-adjustment algorithms.

### Limitations of the Study

The VAuto algorithm has not been previously validated, and its specific role in the panorama of the different ventilation options for the treatment of OSA patients has not been clarified by the same manufacturer. The titration protocol of VAuto was necessarily empiric using, on the one hand, the experimental data obtained during the CPAP titration procedure and, on the other hand, the experience of our team.

Our study tried to cope with an emergency caused by the Philips recall in which patients were bewildered and worried about the need to continue PAP therapy. Nonetheless, we have demonstrated a substantial equivalence between the two devices. However, the adherence and tolerance data about the VAuto mode are limited to the first days of use.

## 5. Conclusions

The study suggests that VAuto is a viable and well-tolerated option for the treatment of patients with OSA and that the switching from two different auto-adjusting bilevel devices can be easily and successfully made without significant changes in the operating setting.

## Figures and Tables

**Figure 1 jcm-11-03157-f001:**
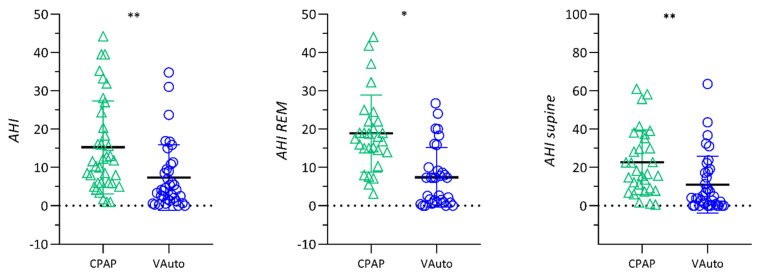
Chart of comparison between residual apnea-hypopnea index (AHI), AHI in REM sleep and AHI in supine sleeping position in Group A. * *p* < 0.05. ** *p* < 0.001.

**Figure 2 jcm-11-03157-f002:**
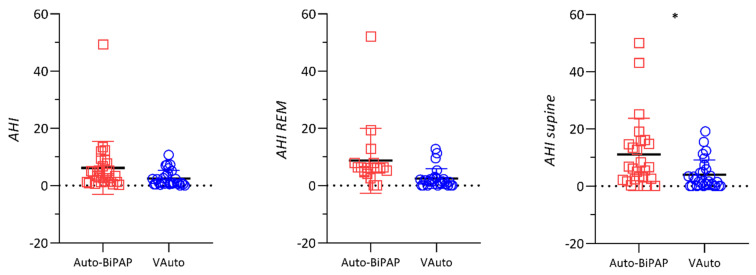
Dot plots represent the comparison of residual apnea-hypopnea index (AHI), AHI in REM sleep and AHI in supine sleeping position between treatment with Auto-BiPAP and VAuto (Group B). * *p* < 0.05.

**Figure 3 jcm-11-03157-f003:**
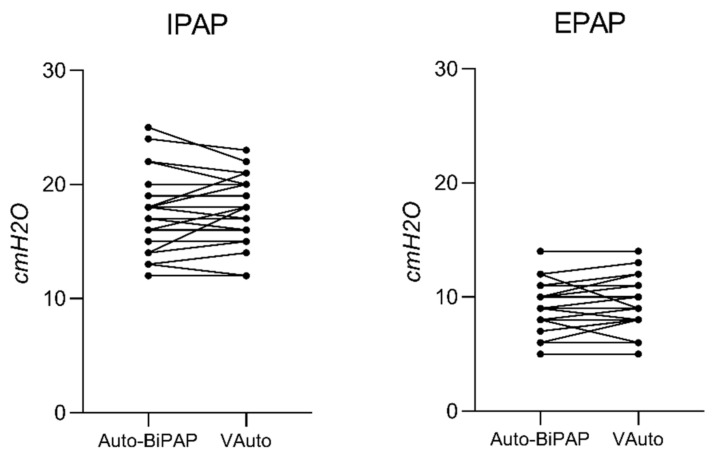
Setting of the therapeutic pressures (IPAP and EPAP) of two auto-adjusting bilevel devices (Auto-BiPAP vs. VAuto) compared in the study.

**Table 1 jcm-11-03157-t001:** Baseline characteristics of population in the study.

	Total (*N =* 64)
** *Demographic* **	
**Gender, % male**	70%
**Age, years**	67.0 ± 9.5
**Height, mt**	1.7 ± 0.1
**Weight, kg**	98.1 ± 18.6
**BMI, kg/m^2^**	34.2 ± 6.6
** *Comorbidities* **	
**CHD, %**	34%
**Atrial Fibrillation, %**	17%
**Hypertension, %**	80%
**COPD, %**	22%
**Cerebrovascular Disease, %**	25%
**CKD, %**	16%
**Diabetes, %**	25%
**Endocrinological Disorder, %**	23%
**Dyslipidemia, %**	39%
**GERD, %**	27%
**Anxiety-Depression, %**	6%
**Malignancy, %**	11%
**Other, %**	67%
** *Sleep Data* **	
**TST, min**	342.7 ± 52.7
**SE, %**	81.6 ± 11.4
**N1, %**	13.6 ± 10.3
**N2, %**	35.8 ± 13.0
**N3, %**	28.1 ± 13.5
**REM, %**	22.4 ± 7.7
**Arousals index, events·h^−1^**	49.2 ± 21.8
**AHI, events·h^−1^**	47.4 ± 21.9
**AHI REM, events·h^−1^**	55.5 ± 19.7
**AHI supine, events·h^−1^**	60.8 ± 26.1
**ODI, events·h^−1^**	38.4 ± 20.8
**T_90_, %**	28.6 ± 26.0
**Mean SaO_2_, %**	91.3 ± 2.7
**Nadir, %**	75.9 ± 8.6
**ESS, points**	7.2 ± 5.3

Numerical data are expressed as mean ± SD, while categorical as percentage. Abbreviations: AHI= apnea–hypopnea index; BMI = body mass index; CHD = chronic heart disease; CKD = chronic kidney disease; COPD = chronic obstructive pulmonary disease; ESS = Epworth sleepiness scale; GERD = gastroesophageal reflux disease; ODI = oxygen desaturation index; SE = sleep efficiency; T_90_ = sleep time with oxygen saturation <90%; TST = total sleep time.

**Table 2 jcm-11-03157-t002:** Demographic and clinical characteristics of two groups examined.

	Group A (*N =* 36)	Group B (*N =* 28)	*p*
** *Demographic* **			
**Gender, % male**	67%	75%	0.469
**Age, years**	67.8 ± 9.6	65.8 ± 9.5	0.412
**Height, mt**	1.7 ± 0.1	1.7 ± 0.1	0.642
**Weight, kg**	100.0 ± 18.3	95.6 ± 19.0	0.352
**BMI, kg/m^2^**	34.6 ± 6.1	33.7 ± 7.3	0.595
** *Comorbidities* **			
**CHD, %**	44%	21%	0.054
**Atrial Fibrillation, %**	22%	11%	0.226
**Hypertension, %**	75%	86%	0.29
**COPD, %**	22%	21%	0.939
**Cerebrovascular Disease, %**	28%	21%	0.56
**CKD, %**	17%	14%	0.794
**Diabetes, %**	33%	14%	0.8
**Endocrinological Disorder, %**	28%	18%	0.352
**Dyslipidemia, %**	42%	36%	0.628
**GERD, %**	22%	32%	0.327
**Anxiety-Depression, %**	3%	11%	0.193
**Malignancy, %**	14%	7%	0.391
**Other, %**	64%	71%	0.294

Numerical data are expressed as mean ± SD, while categorical as percentage. Abbreviations: BMI= body mass index; CHD= chronic heart disease; CKD= chronic kidney disease; COPD= chronic obstructive pulmonary disease; GERD= gastroesophageal reflux disease.

**Table 3 jcm-11-03157-t003:** Comparison of sleep data in Group A (CPAP vs. VAuto) and in Group B (Auto-BiPAP vs. VAuto).

	Group A (*N =* 36)			Group B (*N =* 28)		
	CPAP	VAuto	*p*	Auto-BiPAP	VAuto	*p*
**TST, min**	319.5 ± 62.7	290.9 ± 89.7	0.3	302.2 ± 86.3	338.3 ± 93.6	0.686
**SE, %**	73.7 ± 14.9	71.9 ± 14.9	0.541	79.6 ± 12.8	72.7 ± 16.0	0.048
**N1, %**	11.1 ± 5.3	14.4 ± 14.3	0.675	11.4 ± 9.9	17.7 ± 14.3	0.035
**N2, %**	35.1 ± 8.6	37.9 ± 9.6	0.269	37.7 ± 12.5	35.0 ± 11.2	0.255
**N3, %**	28.6 ± 10.9	27.5 ± 9.1	0.819	31.2 ± 14.4	26.3 ± 13.9	0.06
**REM, %**	24.9 ± 9.2	20.1 ± 11.1	0.043	19.5 ± 10.5	20.9 ± 10.2	0.81
**Arousals index, events·h^−1^**	22.0 ± 11.2	10.5 ± 9.1	<0.001	13.7 ± 9.6	5.4 ± 6.0	0.001
**AHI, events·h^−1^**	15.2 ± 12.1	7.4 ± 8.5	<0.001	6.1 ± 9.2	2.5 ± 2.8	0.057
**CAHI, events·h^−1^**	5.0 ± 8.0	4.1 ± 8.1	0.54	2.0 ± 2.5	0.9 ± 1.8	0.036
**OAHI, events·h^−1^**	10.2 ± 10.4	3.3 ± 4.3	<0.001	4.1 ± 8.4	1.6 ± 1.8	0.145
**AHI REM, events·h^−1^**	19.5 ± 12.5	7.4 ± 8.4	0.002	9.8 ± 13.7	2.5 ± 3.4	0.071
**AHI supine, events·h^−1^**	22.7 ± 17.9	11.0 ± 14.8	<0.001	11.1 ± 12.6	3.9 ± 5.1	0.016
**ODI, events·h^−1^**	15.2 ± 8.8	9.4 ± 8.9	0.029	8.5 ± 10.4	2.6 ± 3.0	0.009
**T_90_, %**	15.4 ± 27.7	7.3 ± 12.2	0.097	3.5 ± 8.1	0.7 ± 3.0	0.086
**Mean SaO_2_, %**	92.4 ± 3.0	93.2 ± 1.8	0.16	94.0 ± 1.8	94.8 ± 1.3	0.032
**Nadir, %**	83.3 ± 6.9	86.4 ± 4.6	0.026	86.4 ± 5.5	89.3 ± 3.2	0.009

All data are expressed as mean ± SD. Abbreviations: AHI = apnea–hypopnea index; CAHI = central apnea–hypopnea index; OAHI = obstructive apnea–hypopnea index; ODI = oxygen desaturation index; SE = sleep efficency; T_90_ = sleep time with oxygen saturation <90%; TST = total sleep time.

## Data Availability

The data that support the findings of this study are available from the corresponding author, upon reasonable request.

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
