# Peer review of "Switch of Nocturnal Non-Invasive Positive Pressure Ventilation (NPPV) in Obstructive Sleep Apnea (OSA)"

_jcm, 2022, doi:10.3390/jcm11113157_

Round 1
Reviewer 1 Report
Thank you for inviting me to review this manuscript on hot topic: Switch of nocturnal non-invasive positive pressure ventilation (NPPV) in obstructive sleep apnea. The manuscript is well written, and the results are very exciting and convincing.
I can propose several points:
1. Please present data on comorbidities in both groups
2. What is your opinion on Baveno classification, please add it in discussions, or if you have data it can be exciting results
3. Please present more limitations of your study
Author Response
Dear Referee,
first of all, thank you very much for your valuable comments.
- Data about comorbidities are now reported.
- Baveno classification has not been yet adopted in our laboratory as a standard procedure (we use it only for specific research purposes) since, up to now, this multidisciplinary score has not been validated. On the other hand, the present study was not designed for this specific aim.
- We included a limitation of the study paragraph in the discussion section accordingly to your suggestion.
Reviewer 2 Report
Important article as it is clinically current and scientifically relevant. However, it becomes crucial to better clarify the method of measuring pressures within each group, after randomization.
I write that, because in the present form i did't understand how had been made the transition from cpap to auto-binivel by resmed.
Had been lab pressure determination of the CPAP.
So for me is necessary a better clarification.
Author Response
Dear Referee,
first of all, thank you very much for your valuable comments.
We agree with you that the VAuto titration protocol was no adequately described in the first version of our manuscript. Now, in the present version we clarified that the titration protocol was made during a standard full in-lab polysomnography.
In the revised version of the manuscript the VAuto titration protocol was described separately for Group A and Group B patients.
